# Research on the relationship between CEO career variety, digital knowledge base extension, and digital transformation in the context of digital merger and acquisition: The case of China's new generation of information technology firms

**Hongyang Li** [1]*, **Xu Yang** [1], **Mingming Meng** [2]

1 School of Economics and Management, Harbin Engineering University, Harbin, China, 2 School of Economics and Management, Beijing Forestry University, Beijing, China

☯ These authors contributed equally to this work.

* lhy_heu@hrbeu.edu.cn

## Abstract

This study examines the relationship between CEO career variety, digital knowledge base extension, and digital transformation in a digital M&A context. An empirical test was conducted using regression analysis with the digital M&A events of the new generation of information technology firms in China as the research sample. The results reveal that CEO career variety has a positive effect on digital transformation in the digital M&A context and that digital knowledge-base extension plays a mediating role. Moreover, the heterogeneity impact analysis indicated that the moderating effects of geographical distance, knowledge disparity, and cultural difference between target and acquirer firms on the above relationships vary greatly: geographical distance has a negative moderating effect, cultural difference has a positive moderating effect, and the moderating effects of both geographical distance and cultural difference are realized through mediating effects, but none of the moderating effects of knowledge disparity are significant.

## 1 Introduction

In recent years, artificial intelligence (AI), blockchain, cloud computing, big data, and other underlying digital technologies have driven the emergence and development of the digital economy, profoundly affecting the industrial structure system and economic growth patterns. Under the role of digital technology empowerment, digital transformation (DT) has become the key to industrial survival and development in the digital economy era. Digital technology is a combination of information, computing, communication, and connectivity technologies [1], and its core technologies (e.g., cloud computing and big data analysis) belong to the category of new generation of information technology (NGIT). Therefore, the NGIT industry has

**Data Availability Statement:** All relevant data are within the paper and its Supporting Information files.

**Funding:** This work was supported by the National Natural Science Foundation of China [grant number 72274044]. The funders had no role in study design, data collection and analysis, decision to publish, or preparation of the manuscript.

**Competing interests:** The authors have declared that no competing interests exist.

an inherent technical advantage in implementing DT. Since China listed NGIT as a strategic and emerging industry in the "12th Five-Year Plan," China's NGIT industry has been developing rapidly. The Central Cyberspace Affairs Commission issued the "14th Five-Year Plan for National Informatization," which pointed out that the NGIT industry shoulders the essential tasks of promoting 1) the optimization and upgrading of traditional industries, 2) the convergence of informatization and industrialization, and 3) the development of intelligent manufacturing. Thus, whether the NGIT industry can take the lead in advancing DT becomes critical.

DT is the process of updating digital resources to improve business, enhance efficiency, transform organizational structures, and reshape innovation models [2, 3]. Although the current development of China's digital economy is robust and new business formats and patterns related to NGIT firms are constantly emerging, many NGIT firms still face difficulties in the actual DT process. From a theoretical point of view, the research and development (R&D) process of internal digital knowledge is often marked by path dependence. Moreover, it is not only time-consuming but also risky to rely on internal innovation when one's digital knowledge base is immature. From a realistic perspective, NGIT firms generally encounter many problems such as unclear digital technology implementation paths, poor digital strategies, and shortages of highly educated digital talent. For this reason, numerous NGIT firms have started to turn their attention to outside firms by acquiring digital companies that fit their DT strategy to quickly and directly obtain digital assets to enhance transformation capabilities and control transformation costs. This type of strategic acquisition targeting digital economy firms is called "digital mergers and acquisitions" (M&A) [1]. However, while digital M&A has become an important strategic choice for an increasing number of NGIT firms [4], compared to the burgeoning digital M&A practice, digital M&A theoretical research is relatively lagging and weak. While there are numerous studies on DT in non-M&A settings [2, 3], research into DT in the context of digital M&A has received little attention, and the specific paths and mechanisms promoting DT in digital M&A contexts remain unclear.

As an influential decision-maker and executive, the chief executive officer (CEO) is an important influencing factor in a firm's DT [5]. As modern DT is marked by a complex and unstable internal and external environment, a high level of CEO digital awareness and digital leadership coupled with a keen insight into digital trends and industry development are required; these differences in CEO capabilities are important factors in the widening DT gap [6]. In this situation, compound talent meets firms' requirements for overall CEO quality and tends to positively impact corporate innovation investment, corporate M&A, and other behaviors [7]. Wealthy career experience is an important way to shape compound talent [8]. For example, Ren Zhengfei, the founder of China's Huawei, previously served in military and constructional engineering units and Shenzhen Nanhai Petroleum Industry Co., Ltd., which enabled him to develop Huawei into a world-renowned NGIT firm.

In the digital M&A context, this study considers DT a type of strategic decision-making and examines the influence of CEO career variety on DT. Considering the importance of digital knowledge-based extension (DKBE) in the digital M&A process to achieve DT [9], DKBE is employed as a mediating variable in the relationship between CEO career variety and DT. This study screens NGIT firms from Chinese A-share listed firms and collects their digital M&A events as the research sample. It empirically examines the mediating relationship between CEO career variety, DKBE, and DT using a regression analysis method and the moderating effects of geographical distance, knowledge disparity, and cultural differences on the above relationships are also explored in a heterogeneity analysis.

The analysis of the results reveals several theoretical implications. First, it adds value to the literature on CEO career variety and DT in the emerging digital M&A research context. The

study of digital M&A and DT is still in its infancy; while previous literature has addressed the impact of executives and DT [10, 11], few scholars have focused on the relationship between CEO career experience and DT. In response to firms' actual demand for composite talent in the digital economy, this study clarifies the positive impact of CEO career variety on DT in the digital M&A context. This study provides an empirical basis for firms to utilize the dual advantages of CEO capabilities and resources to promote DT. Second, the study reveals that there is a bridging role between CEO career variety and DT, providing new insights for digital knowledge-based research. Notably, it is worth noting that this mechanism holds only in the digital M&A context. Although previous technology M&A studies focused on knowledge bases [12, 13], detailed research on how DKBE is lacking. This study clarifies the mechanisms of CEO career variety, DKBE, and DT in the digital M&A context, opens a "black box" for DT, and complements firms' DKBE problems. Finally, this study explores in-depth the factors that influence the relationship between CEO career variety and DT from the perspective of M&A parties' heterogeneity. By exploring the moderating effects of geographic distance, knowledge disparity, and cultural differences, this study deepens the understanding of the different boundary effects of geographic, knowledge, and cultural factors between M&A parties in the digital M&A context, enriches theoretical research in the digital M&A field, and inspires NGIT firms to pay more attention to the target selection issue.

The remainder of the paper is structured as follows: Section 2 reviews the existing literature and derives the hypotheses. Section 3 explains the research design. Section 4 provides the empirical results, robustness tests, and heterogeneity analysis results. Section 5 presents the discussion. Finally, Section 6 presents the conclusions and policy suggestion.

## 2 Literature review and hypotheses

### 2.1 Literature review

The concept of digital M&A s is based on early research on technological M&A in the digital economy. Since Ahuja and Katila [14] introduced the concept of technology M&A in 2001, technology M&A research has developed rapidly, research on digital M&A is still in its infancy. Although some scholars have focused on general M&A research in the context of the digital economy [15–17], their focus differs somewhat from digital M&A. The term digital M&A was first introduced in a research note published by Bain and Company. Subsequently, the idea of "firm DT can be achieved through digital M&A" began to attract scholars' attention [18, 19]. However, to date, studies have focused only on the theoretical level of digital M&A. A search of the Web of Science database reveals only a handful of empirical studies on digital M&A. One such study is Hanelt et al. [1], who through systematic elaboration, theoretical conceptualization, and empirical testing of digital M&A, argued that digital M&A helps promote digital innovation and boosts firm performance. Additionally, Zhou et al. [9], using data of Chinese listed firms as a sample, verified the positive relationship between digital M&A and innovation performance. Yu and Yan [20] argued that digital finance development promotes the implementation of digital M&A by firms. Finally, Tang et al. [21] focused on the positive impact of digital M&A on the market value of Chinese listed firms. However, there is still a lack of research on how to achieve the goal of firm DT through digital M&A.

In the field of influencing factors of firm DT, early scholars focused on technical aspects such as digital resources and digital-related abilities [3, 5]. However, as the internal obstacles to DT within firms continue to increase, the organizational element is gradually gaining attention. Scholars have investigated the role of typical organizational factors such as organizational structure, organizational culture, and organizational governance can influence firm DT [3, 22], and in this process, the role of managers has gradually emerged. Previous studies have

delved into two aspects: manager characteristics and manager abilities, and found that both psychological characteristics, cognitive structures, overseas backgrounds [10, 11, 23], and digital literacy, digital self-efficacy, and management abilities [24–26] have significant impacts on DT. Additionally, the impact of the Chief Digital Officer, a specialized senior management position, on DT has also received attention [27, 28].

As the main decision-makers in the daily management of firms, CEOs have been the focus of attention in business and strategic management research. Traditional agency theory focuses on how to guide managers to make "pareto optimality" decisions to promote sustainable development, but the implied premise of managerial homogeneity does not easily match the real situation and is gaining increasingly more attention from scholars. Based on the upper echelons theory, CEOs, as the core of management, have career experiences that influence their cognitive and behavioral patterns, which, in turn, influences firms' behavior. Previous research on CEO career variety has focused on the economic impacts of single specific career experiences such as military, R&D, and financial experience on firms [29–31]. However, different career types interact with each other to shape the management style of the CEO. Some studies have found that CEOs employ a combination of skills learned throughout their careers when making corporate decisions, and that CEOs with a wide range of career experiences are usually more strained, boundary-spanning, innovative, and adventurous [8, 32]. However, current research has centered solely on the impact of CEO career variety on outcomes such as CEO compensation, investment decisions, and board tenure [8, 33, 34], and there is a lack of in-depth exploration into the consequences resulting from CEO career variety.

## 2.2 Hypotheses proposed

**CEO career variety and firm DT.**   The upper echelons theory, first proposed by Hambrick and Mason [35], argues that management's values and cognitive abilities and different executive characteristics significantly impact decision-making and execution in a firm. This study refers to Crossland et al.'s [7] concept of CEO career variety; That is, the array of distinct professional and institutional experiences that an executive has gained prior to becoming CEO. Considering that DT in the digital M&A context serves as an important growth strategy for firms, CEOs play a major role in decision-making and leadership. This implies that a firm's DT in the digital M&A context is also influenced by the CEO's career variety [11]. Thus, according to the upper echelons theory, this study conducts a theoretical analysis from the perspective of imprinting and resource effects.

From the imprinting effect perspective, CEOs' work experiences in different organizations or environments influence their management thinking and decision preferences through cognitive and competence imprinting [36]. CEOs with diverse career experiences are more cognitively aware of DT and more capable of making relatively better DT decisions, thus enhancing the degree of firm DT. Achieving DT through digital M&A is both an opportunity and a challenge for firms. Firms are not only under pressure to integrate digital M&A, but also face serious problems such as unclear transformation goals and uncertainty about the direction in the process of DT. It has been shown that CEOs with varied careers have broader knowledge, stronger overall skills, and the capacity to identify major opportunities and challenges in a firm [33]. Therefore, when faced with DT opportunities, CEOs with greater career variety have more advanced cognitive skills and awareness than those with less career variety and are more likely to recognize the importance of DT to their organizations. They can seize major opportunities for digitalization and organizational restructuring, that enhance the value of their firms.

From the resource effect perspective, resource-based theory suggests that CEOs, by virtue of their social connections, can become valuable social resources by employing a greater range

of information sources, improving information quality, and alleviating information asymmetry [33, 37]. The embeddedness theory of new economic sociology also states that the economic behavior of CEOs in society is embedded in their social network ties, thus forming social capital prototypes [38]. As an important strategic decision, DT in the context of digital M&A has resource-dependent attributes, and firms need strong social capital to support their DT efforts, which are both high risk and high reward. Managers in different firms and industries inevitably expand the boundaries of interpersonal interactions and enrich their social relationships [39]. Abundant career experience can enable managers to accumulate many quality resources, including capital, talent, and knowledge. CEOs with career variety can leverage their rich career experience to bring in external social resources, especially financial ones, to better support their firms' DT. Combining these two perspectives led us to propose the following hypotheses:

H1: CEO career variety positively impacts DT in the context of digital M&A.

**CEO career variety and DKBE.** A digital knowledge base is defined as the sum of all explicit and tacit aspects of a firm's digital knowledge, including digitally relevant information, knowledge, and capabilities that inventors utilize to find innovative solutions [40, 41]. The fundamental purpose of digital M&A is to acquire the digital knowledge of the target firm, gradually merging the digital knowledge bases of both firms and showing a trend of knowledge transfer from the target firm to the acquirer firm [1]. However, due to the inherent "stickiness" of knowledge, it is difficult to realize knowledge transfer through M&A [13]. Knowledge transfer is a complicated process involving bilateral interactions between the sender and receiver of knowledge. Accordingly, the motivation, capability, and opportunity of both target and acquirer firms directly influence effective knowledge transfer [42]. The CEO plays a major decision-making and driving role in the knowledge transfer process, and CEO career variety has a significant impact on DKBE.

Based on the imprinting effect perspective, CEOs with different career experiences interact with each other to form their own unique imprint, which significantly affects corporate decision-making and execution. CEOs with professional variety can effectively reduce semantic bias in the digital M&A integration process, facilitate idea sharing and understanding, more easily maintain a good relationship with the target firm, and promote mutual cooperation between the target and acquirer firm [36, 43], and digital knowledge transfer. Considering the information asymmetry in the digital M&A process, CEOs with diverse careers can efficiently obtain more non-redundant information by virtue of their rich career experience, thus alleviating information asymmetry in corporate decision-making and facilitating digital knowledge transfer. Therefore, CEOs with career variety are better able to eliminate internal differences and gain the full trust and support of both firms, thus accelerating DKBE.

From the resource effect perspective, CEOs with career variety and who have worked in several different functions, businesses, geographies, and organizations can acquire more social resources and social network relationships for the firm [39]. As auxiliary resources, these social resources can effectively facilitate digital knowledge transfer and strengthen the construction of their own digital knowledge bases. Simultaneously, as an informal institutional arrangement, the resource allocation effect of social networks can help firms obtain scarce resources [44], thus facilitating DKBE. In addition, diverse career experiences endow CEOs with diverse and complex management knowledge bases and a pioneering sense of external perception, leading to a greater information advantage in decision-making, which significantly improves the quality of digital M&A decisions and facilitates DKBE. The following research hypothesis is proposed by combining the analyses of these two perspectives:

H2: CEO career variety has a positive effect on DKBE in the context of digital M&A.

**Mediating effects of DKBE.** From the above analysis, it is clear that based on the upper echelons and imprinting theories, the imprinting effect formed by the rich professional experience of CEOs not only helps firms promote digital knowledge transfer and expand their digital knowledge base but also enables them to better grasp the opportunities for DT and promote the degree of firm DT. In contrast, based on the resource-based and embeddedness theories, the resource effect of CEO's rich past professional experience helps improve the quality of decision-making, accelerates the construction of a digital knowledge base, and helps alleviate the shortage of resources faced by firms in DT. Thus, it appears that CEO career variety in the digital M&A context can contribute to both DKBE and DT.

The construction of a digital knowledge base in the digital M&A process is crucial for achieving DT goals. A digital knowledge base not only enables firms to apply and combine this knowledge directly but also increases the acceptance of new external knowledge. Building a digital knowledge base enables firms to absorb and utilize external digital knowledge [1], which facilitates the DT process [45]. In addition, Zhou et al. [9] point out that in a digital M&A scenario, acquiring the digital knowledge of the target firm can effectively facilitate firm DT. Therefore, CEO career variety in digital M&A scenarios can facilitate DKBE and thus increase the degree of DT. This demonstrates that DKBE serves as a link between CEO career variety and DT. Based on the above analysis, the following hypotheses are proposed:

H3: DKBE mediates the relationship between CEO career variety and DT in digital M&A.

## 3 Research design

### 3.1 Research sample

First, based on the Guidance List of Key Products and Services in Strategic Emerging Industries (2016 edition) published by China's National Development and Reform Commission, we first matched the key directions and refined sub-directions of the NGIT industry with the concept board of listed firms in the Wind database and then defined listed firms that fit the business scope as NGIT listed firms. Second, we collected the equity M&A events for such firms from 2013 to 2017. Samples with too small quantities, failed M&A, and missing data were excluded. Finally, referring to the criteria for defining digital M&A proposed by Hanelt et al. [1], M&A events whose M&A targets are digital economy-type firms were screened out and defined as digital M&A events in this study. The "Statistical Classification of the Digital Economy and its Core Industries" (2021), issued by the National Bureau of Statistics (NBS), was used as the standard for judging digital economy-type firms. Specifically, the industry codes, business scope, and business description of the target firms were collected through Wind, Beijing Tianyancha Technology Co., Ltd. and Qichacha Tec Co., Ltd., and the industry code information of the target firms was matched with the national economic industry code of the statistical classification. The business scope and description information were matched with the industry description of the statistical classification. If the information about the target firm matched the statistical classification, the target firm was deemed a digital economy-type firm, and the sample was considered a digital M&A event. A total of 415 digital M&A were identified using the screening steps described above.

### 3.2 Measurement

**Independent variable.** The DT variable reflects the firm's degree of transformation through digital technology. Related studies typically use text mining and statistical word

frequencies to measure this variable [46]. Referring to Chen [47] and based on the China Securities Market and Accounting Research (CSMAR) database on digital-related measures of listed firms, we illustrated the degree of firm DT in five dimensions (AI technology, blockchain, cloud computing, big data, and digital technology applications) and mined the frequency of the corresponding text words in the annual reports of listed firms. The word frequencies of these five dimensions were summed and logarithmically processed after adding one to depict the level of firm DT. At the same time, considering that there is a certain lag period from the role of CEO career variety to DT in the digital M&A context, the DT data were treated with a lag of four years.

**Dependent variable.** Referring to Custódio et al. [8] and Crossland et al. [7], CEO career variety was measured by constructing a CEO career experience richness index. The following five aspects were considered in the index construction process: career type, number of firms, academic experience, financial institutions, and geographical type.

Career type is a variable measured by the aggregate number of different functions experienced by a CEO throughout his/her career. Considering the data sources, and based on the studies of Crossland et al. [7] and Schmid and Mitterreiter [34], we divided the occupation types into a total of nine types: production, R&D, design, human resources, management, marketing, financial market profession, financial management profession, and legal. Each occupation type was not counted twice; those who worked in the same occupation type in multiple firms were counted only once, and the data were obtained from the CSMAR database.

The number of firms was measured by the aggregate number of different firms in which a CEO has worked throughout his/her career. CEOs who have worked in multiple firms have a better understanding of how different firms operate and thus exhibit stronger management abilities. The number of firms was calibrated according to Crossland et al.'s [7] method. Multiple firms experienced by a CEO were considered the same if they belonged to a subsidiary of a business group or had a parent and subsidiary corporation relationship. The data for this variable were obtained by manually querying the Wind database for CEO biographies.

Academic experience was measured using the aggregate number of academic institutions the CEO has experienced throughout his/her career. CEOs with multiple academic backgrounds have relatively rich academic resources and higher social statuses. The statistical categories included teaching at universities, serving at research institutions, and conducting research at associations. The data were obtained from the CSMAR database.

Financial institutions was measured as the aggregate value of the number of different financial institutions that the CEO has experienced throughout his/her career. Referring to the CSMAR database's financial institution classification criteria, the statistical scope includes supervision departments, policy banks, commercial banks, insurance companies, security companies, fund management companies, securities registration and settlement companies, futures companies, investment banks, trust companies, investment management companies, and stock exchange. The data were obtained from the CSMAR database.

Geographical type was measured the different types of geographical experience a CEO has had throughout his/her career. A CEO with overseas experience was assigned a value of one if he/she has worked or studied abroad, and zero otherwise. A CEO with overseas experience is likely to have a higher level of vision and more advanced ideas in firm decision-making owing to exposure to different cultures. Data were obtained from the CSMAR database.

To avoid the multicollinearity issue, we employed the factor analysis method to reduce the dimensionality of the aforementioned indicators and computed the comprehensive score for the CEO career experience richness index through weighting. The first step involved standardizing the five indicators and calculating the KMO value as well as the Bartlett test of sphericity to determine the applicability of the factor analysis method. Subsequently, in the second step,

we extracted the common factor based on the criterion of an eigenvalue greater than one and calculated the score of the common factor in different samples. Finally, in the third step, we constructed a comprehensive evaluation function that relied on the common factor score and the variance contribution rate of this factor to obtain a comprehensive score. The specific calculation formula is as follows:

$$Index_{ti} = (\beta_{i1}F_{i1} + \beta_{i2}F_{i2} + \beta_{i3}F_{i3} + \cdots\cdots + \beta_{ij}F_{ij})/(\beta_{i1} + \beta_{i2} + \cdots\cdots + \beta_{ij}) \tag{1}$$

Where $Index_{ti}$ represents the comprehensive score of the CEO career experience richness index for the ith sample company in year t; $F_{ij}$ represents the extracted common factor; and $\beta_{ij}$ represents the variance contribution rate of the j-th common factor for the i-th sample company.

After calculation, the KMO value of the samples is 0.716, and the chi-square value of Bartlett's sphericity is 270.465, which is significant at the level of 1%, indicating that the samples are suitable for factor analysis. The common factor extraction degrees are all above 0.5, and the extraction degree of the original variables is relatively high. The common factors are extracted according to the standard that the characteristic root is greater than 1, and after rotation by the maximum variance method, the cumulative variance contribution rate is 62.044%, which can better represent most information of the original variables. Finally, the comprehensive score of CEO's career experience richness index is calculated based on Formula (1).

**Mediating variables.** DKBE refers to the degree of expansion of the digital knowledge base of acquirer firms after a digital M&A. Knowledge bases are typically measured using patent quantities [48]. Considering that invention patents have the characteristics of originality, exclusivity, and exclusivity and that the invention patent examination procedure is more stringent and requires more innovative products and technologies, this we used the number of digital invention patent applications to measure the size of the digital knowledge base. There is a lag period for DKBE relative to digital M&A. Referring to Stiebale's [49] three-year lag period setting, we used the total number of digital invention patent applications in the three years after the digital M&A to represent the degree of expansion of the digital knowledge base of acquirer firms after a digital M&A [1].

The screening practices for digital invention patents were as follows. First, all patent IPC classification numbers of acquirer firms were searched and collected through the State Intellectual Property Office. Second, "the Reference Relationship Table Between International Patent Classification and Industrial Classification for National Economic Activities" (2018), which was issued by China National Intellectual Property Administration (CNIPA), was matched with the "Statistical Classification of Digital Economy and its Core Industries" (2021) promulgated by the National Bureau of Statistics to screen out the categories of the National Economic Industry Classification that are in line with the digital economy industry. Finally, all patent IPC classification numbers of each acquirer firm were compared with the National Economic Industry Classification, and invention patents that fit into the digital economy industry category were defined as digital invention patents.

**Control variables.** The control variables were selected from the perspective of M&A parties' firm differences and acquirer firm heterogeneity as follows:

*Geographical distance.* Referring to Sears [50], the geographic distance indicator was measured as the geographic straight-line distance between the locations of the target and acquirer firms.

*Knowledge disparity.* The knowledge disparity between target and acquirer firms is reflected in the difference in knowledge-based stocks. Considering that invention patents are more innovative than utility model and design patents, the ratio of the total number of invention

patents granted between the target firm and the acquirer firm in the five years prior to the M&A was used as a measurement indicator. The larger the ratio, the larger the disparity between the target firm's technological capability and that of the acquirer. Moreover, the closer the ratio is to one, the closer the technological capabilities of both parties.

*Cultural differences.* Many sociological studies have concluded that language has both social and cognitive functions and is an important carrier of regional culture [51]. Dialects contain different patterns of thinking and behavior, and organizations or individuals in a particular dialect environment are implicitly influenced by regional cultures. Therefore, this study used dialect differences between the regions where the target and acquirer firms are located to measure cultural differences [52]. Referring to the "Language Atlas of China" (2012), we counted the dialectal regions of the locations of M&A parties. If the locations of either party did not belong to the same dialectal region, it was recorded as three. If both locations belonged to the same dialectal region but not the same dialectal area, it was recorded as two. If the location of both parties belonged to the same dialectal area but not the same subdialect, it was recorded as one. If both locations belonged to the same subdialect, it was recorded as zero. This provides us with the degree of dialectal difference between M&A parties, where the larger the value, the larger the difference.

*Firm size.* Considering that an M&A event itself is affected by the size of the acquirer, creating an acquirer-size effect, it is necessary to identify possible biases in the results due to firm size. Firm size was measured as the natural logarithm of the acquirer's total assets in the year before the merger.

*Leverage.* This study used the ratio of the total liabilities divided by the total assets of the acquirer firm in the year prior to the M&A to measure leverage.

*Growth.* Growth was measured using the growth rate of the acquirer's operating income in the year prior to the M&A.

*M&A experience.* Referring to Zhou et al. [9], the combined number of times the acquirer firm went to acquire the target firm in the five years prior to the M&A was counted and used to measure the firm's M&A experience.

Table 1 presents the names and measurements of each variable.

## 3.3 Model

The stepwise testing procedure proposed by Baron and Kenny [53] is currently regarded as the most popular method for testing the mediating effect. Baron and Kenny [53] argue that the test of the mediating effect should involve three steps: Firstly, testing the total effect of the independent variable on the dependent variable (i.e., whether β1 in Eq 2 is significant); Secondly, testing the effect of the independent variable on the proposed mediator (i.e., whether β1 in Eq 3 is significant); Thirdly, testing the effect of the proposed mediator on the dependent variable controlling for the independent variable (i.e., whether β2 in Eq 4 is significant). Hayes [54] also refers to this process as the causal steps approach or joint significance test.

$$DT = \beta_0 + \beta_1 CEO\ career\ variety + \beta_2 Contorl + \mu \tag{2}$$

$$DKBE = \beta_0 + \beta_1 CEO\ career\ variety + \beta_2 Contorl + \mu \tag{3}$$

$$DT = \beta_0 + \beta_1 CEO\ career\ variety + \beta_2 DKBE + \beta_3 Contorl + \mu \tag{4}$$

Where $Control_{t-1}$ represents six control variables. $\beta_0$–$\beta_6$ represent the coefficients of each variable, and μ represents the random error term. Subsequent empirical tests were conducted by using the STATA software.

**Table 1. Descriptive statistics of the variables.**

| Variable | Name | Measurement |
|---|---|---|
| Independent variable | DT | Natural logarithm of the total value of digitized keyword terms frequency in the annual report of acquirer firms in the fourth year after the digital M&A |
| Dependent variable | CEO career variety | The index of CEO career experience richness is constructed from five aspects: career type, number of firms, academic experience, financial institutions, and geographical type |
| Moderator variables | DKBE | The total number of digital invention patent applications in the three years after the digital M&A |
| Control variables | Geographical distance | The geographic straight-line distance between the locations of the target and acquirer firms |
| | Knowledge disparity | The ratio of the number of invention patent applications between the target and acquirer firms five years before the M&A |
| | Cultural difference | The dialect differences between the regions where the target and acquirer firms |
| | Firm size | Natural logarithm of the total assets of the acquirer firms one year before the M&A |
| | Leverage | The ratio of total liabilities to total assets of the acquirer firms one year before the M&A |
| | Growth | The growth rate of operating revenue of the acquirer firms one year before the M&A |
| | M&A experience | The combined number of times the acquirer firm went to acquire the target firm in the five years prior to the M&A |

## 4 Results

### 4.1 Descriptive statistics and correlation analysis

The descriptive statistics for each variable are shown in Table 2. As shown in Table 2, DT has a mean value of 2.837 and a standard deviation (SD) of 1.567, indicating that the degree of DT varies widely among firms. The CEO career variety indicator was obtained using the factor analysis method, which downscales the original five-dimensional indicators to a mean value of zero and a SD of 0.578. In addition, considering the possible problem of multicollinearity among variables, this study conducted a correlation analysis and variance inflation factor (VIF) test for each variable. Table 3 shows that the correlation coefficients of the core variables are all less than 0.3, and the VIFs fall below the standard of 10, indicating the absence of multicollinearity issues. Furthermore, we observed that there is generally no linear correlation between the core variables, which suggests the possibility of non-linear relationships within the data as well as heteroscedasticity issues that merit further investigation in subsequent analyses.

**Table 2. Descriptive statistics of the variables.**

| | Mean | Sd | Min | Max |
|---|---|---|---|---|
| DT | 2.837 | 1.567 | 0 | 6.082 |
| CEO career variety | 0 | 0.578 | -0.742 | 2.317 |
| DKBE | 104.366 | 255.977 | 0 | 2532 |
| Geographical distance | 609.392 | 618.759 | 7.46 | 3206 |
| Knowledge disparity | 0.408 | 1.383 | 0 | 18 |
| Cultural difference | 1.925 | 1.382 | 0 | 3 |
| Firm size | 21.681 | 1.004 | 19.593 | 26.547 |
| Leverage | 0.349 | 0.177 | 0.033 | 0.805 |
| Growth | 0.200 | 0.458 | -0.456 | 5.598 |
| M&A experience | 1.957 | 4.396 | 0 | 39 |

**Table 3. Correlation analysis results.**

| | DT | CEO career variety | DKBE | Geographical distance | Knowledge disparity | Cultural difference | Firm size | Leverage | Growth | M&A experience |
|---|---|---|---|---|---|---|---|---|---|---|
| DT | 1 | | | | | | | | | |
| CEO career variety | 0.049 | 1 | | | | | | | | |
| DKBE | 0.155 *** | 0.034 | 1 | | | | | | | |
| Geographical distance | -0.080 | 0.037 | -0.033 | 1 | | | | | | |
| Knowledge disparity | 0.003 | 0.048 | 0.050 | 0.007 | 1 | | | | | |
| Cultural difference | -0.172*** | 0.052 | -0.074 | 0.715*** | 0.003 | 1 | | | | |
| Firm size | -0.128*** | 0.004 | 0.386*** | 0.008 | 0.060 | 0.009 | 1 | | | |
| Leverage | -0.112** | -0.034 | 0.179*** | -0.060 | -0.012 | -0.121** | 0.553*** | 1 | | |
| Growth | 0.074 | 0.110** | 0.039 | 0.065 | -0.000 | 0.089* | 0.129*** | 0.010 | 1 | |
| M&A experience | -0.047 | -0.117** | 0.131 | 0.022 | 0.032 | -0.004 | 0.419*** | 0.353*** | 0.003 | 1 |
| VIF | — | 1.03 | 1.19 | 2.06 | 1.01 | 2.11 | 1.85 | 1.53 | 1.04 | 1.27 |

Note: *Significant at the 10% level, for two-tailed tests.

**Significant at the 5% level, for two-tailed tests.

***Significant at the 1% level, for two-tailed tests.

## 4.2 Baseline regression results

Based on the correlation analysis presented above, it is found that there is no significant linear relationship between the core variables, and unbalanced panel data is prone to heteroscedasticity. Therefore, we proceeded to the following steps to make a judgment. Firstly, all data were standardized to ensure consistency in the data magnitudes of all variables. Secondly, a White test is conducted to determine the presence of a heteroskedasticity problem in the data based on the significance of coefficients. The White test results show evidence of heteroscedasticity in the data. Finally, we further established that the data had the problem of inter-group heteroscedasticity using the Wald test, which is also a common issue in unbalanced panels. After a comprehensive analysis of the above studies, we selected the feasible generalized least squares (FGLS) model as the most suitable approach for addressing unbalanced panel data while accounting for heteroskedasticity issues. Table 4 presents the regression results, which show that the Wald chi squared values are all significant at the 1% level, indicating that the overall effect and stability of each model is good.

Model 1 shows that CEO career variety has a significant positive effect on DT ($\beta = 0.082$, $p<0.01$), indicating that CEO career variety can promote firm DT degree in digital M&A context, thereby verifying H1. The results of Model 2 show that CEO career variety has a significant positive effect on DKBE ($\beta = 0.040$, $p<0.01$), which indicates that CEO career variety can also help firms rapidly expand their digital knowledge base in the digital M&A context, providing support for H2. The results of Model 3 show that DKBE has a significant positive effect on DT ($\beta = 0.231$, $p<0.01$), while the positive effect of CEO career variety on DT still holds, indicating that there is a partial mediating effect of DKBE between CEO career variety and DT, and H3 holds.

## 4.3 Robustness tests

This study conducted robustness tests from two aspects, namely variable substitution test and endogeneity test.

**Table 4. Empirical test results (N = 415).**

| Variables | 1 | 2 | 3 |
|---|---|---|---|
| | DT | DKBE | DT |
| CEO career variety | 0.082***(4.74) | 0.040***(4.16) | 0.078***(6.55) |
| DKBE | — | — | 0.231***(21.01) |
| Geographical distance | 0.109***(8.17) | 0.044***(4.83) | 0.101***(9.01) |
| Knowledge disparity | 0.019(1.46) | 0.017***(3.13) | 0.019(1.50) |
| Cultural difference | -0.264***(-17.46) | -0.093***(-11.20) | -0.236***(-23.34) |
| Firm size | -0.103***(-12.26) | 0.344***(29.40) | -0.192***(-13.97) |
| Leverage | -0.097***(-9.98) | -0.050***(-7.02) | -0.067***(-4.48) |
| Growth | 0.083***(2.75) | -0.003(-0.34) | 0.121***(3.88) |
| M&A experience | 0.029(1.37) | -0.023(-1.99) | 0.016(0.95) |
| Wald chi2 | 3819.89*** | 1202.38*** | 1400.77*** |

Note: The Z test value is in parentheses.

*Significant at the 10% level, for two-tailed tests.

**Significant at the 5% level, for two-tailed tests.

***Significant at the 1% level, for two-tailed tests.

**Variable substitution test.** In this study, the tests were conducted via the following variable substitutions: (1) changing the dimensionality reduction method of CEO career variety from the factor analysis method to the entropy method, (2) changing the DKBE measure from the original number of numerical invention patent applications to the number of numerical patent applications, and (3) using technicist as a control variable, measured by the total number of technicians of the acquirer in the year before the M&A. Regression analysis was performed based on the original model, the results of which are presented in Table 5. None of the core variables of the model changed substantially and the results remained robust.

**Endogeneity test.** The characteristics of firms that choose digital M&A s may not be the same as those that do not; that is, whether a firm chooses M&A is not a random event [55]. To

**Table 5. Variable substitution test results (N = 415).**

| Variable | 1 | 2 | 3 |
|---|---|---|---|
| | DT | DKBE | DT |
| CEO career variety | 0.142***(10.11) | 0.010***(2.96) | 0.147***(11.20) |
| DKBE | — | — | 0.160***(6.80) |
| Geographical distance | 0.112***(8.31) | -0.029***(-7.20) | 0.111***(7.57) |
| Knowledge disparity | 0.031*(1.72) | 0.015***(6.01) | 0.033**(1.99) |
| Cultural difference | -0.252***(-25.80) | -0.016***(-6.33) | -0.238***(-18.62) |
| Firm size | -0.218***(-14.54) | 0.151***(25.67) | -0.234***(-15.76) |
| Leverage | -0.085***(-7.57) | -0.006(-1.63) | -0.076***(-5.13) |
| Growth | 0.074**(2.38) | -0.042***(-5.27) | 0.077**(2.44) |
| M&A experience | 0.024(1.34) | -0.041***(-16.25) | 0.023(1.21) |
| Technicist | 0.250***(10.53) | 0.478***(24.33) | 0.132***(8.96) |
| Wald chi2 | 3104.88*** | 4668.79*** | 4864.90*** |

Note: The Z test value is in parentheses.

*Significant at the 10% level, for two-tailed tests.

**Significant at the 5% level, for two-tailed tests.

***Significant at the 1% level, for two-tailed tests.

**Table 6. Endogeneity test results (N = 415).**

| Variable | 1 | 2 | 3 |
|---|---|---|---|
| | DT | DKBE | DT |
| CEO career variety | 0.658***(3.42) | 0.039***(3.89) | 0.062***(4.45) |
| DKBE | — | — | 0.229***(13.80) |
| Inverse Mills Ratio | -0.314***(-3.58) | 0.217***(3.97) | -0.356***(-3.28) |
| Geographical distance | 0.127***(8.19) | 0.043***(4.54) | 0.121***(7.13) |
| Knowledge disparity | 0.018(1.41) | 0.015**(2.57) | 0.014(1.08) |
| Cultural difference | -0.266***(-16.08) | -0.098***(-11.82) | -0.236***(-18.89) |
| Firm size | -0.103***(-13.93) | 0.368***(34.27) | -0.175***(-11.19) |
| Leverage | -0.097***(-8.42) | -0.052***(-6.85) | -0.094***(-7.20) |
| Growth | 0.094***(3.63) | -0.003(-0.38) | 0.124***(3.63 |
| M&A experience | 0.034(1.64) | -0.022*(-1.88) | 0.025(1.32) |
| Wald chi2 | 5859.59*** | 1539.59*** | 1129.51*** |

Note: The Z test value is in parentheses.

*Significant at the 10% level, for two-tailed tests.

**Significant at the 5% level, for two-tailed tests.

***Significant at the 1% level, for two-tailed tests.

avoid bias in the estimation results, the Heckman two-stage method is used to test for endogeneity. First, we select a sample of new generation information technology listed companies from 2013 to 2017 and conduct a panel probit with whether the company conducted digital M&A in the current year as the explanatory variables, and a total of eight indicators, including firm size, total asset turnover, total asset return, return on net assets, gearing ratio, growth, number of technicians, and amount of R&D investment in the year before M&A as explanatory variables. The regression, and the Inverse Mills Ratio are calculated. Second, we put the Inverse Mills Ratio into the mediation test model as a control variable. As shown in Table 6, the Inverse Mills Ratio values in the model are significant, indicating that there is a certain selection bias issue in the model. However, after controlling for Inverse Mills Ratio, the core variables in all models do not undergo substantial changes, indicating that the selection bias issue has a relatively small impact on the results and the study findings remain robust.

### 4.4 Heterogeneous impact analysis

Considering that the digital M&A context involves two subjects—the acquirer firm and the target firm—the mediating effect process from CEO career variety, DKBE, and DT are not only influenced by the acquirer firm but also by the target firm; Thus, the boundary effect generated by the difference factors between the target and acquirer firms needs to be considered. Accordingly, in this study, geographical distance, knowledge disparity, and cultural differences between the M&A parties in the original control variables were used as moderating variables to conduct a heterogeneity impact analysis based on the original model. In addition to the original three models, the product terms of explanatory and moderating variables, namely, "CEO career variety x Geographical distance," "CEO career variety x Knowledge disparity," and "CEO career variety x Cultural difference" were added. Table 7 shows the empirical results.

The coefficient of "CEO career variety x Geographic distance" is negative in all three models and significant at the 1% level, indicating that geographic distance has a negative moderating effect on the relationship between CEO career variety and DT. This moderating effect is

**Table 7. Heterogeneous impact analysis results (N = 415).**

| Variable | 1a | 2a | 3a |
|---|---|---|---|
| | DT | DKBE | DT |
| CEO career variety | 0.084**(2.14) | 0.051***(2.75) | 0.025(0.76) |
| DKBE | — | — | 0.228***(17.52) |
| Geographical distance | 0.112***(7.11) | 0.044***(7.69) | 0.089***(5.19) |
| Knowledge disparity | 0.011(0.66) | 0.017**(2.05) | 0.003(0.21) |
| Cultural difference | -0.266***(-15.24) | -0.098***(-19.75) | -0.234***(-12.92) |
| CEO career variety×Geographical distance | -0.084***(-3.14) | -0.055***(-4.72) | -0.080***(-2.71) |
| CEO career variety×Knowledge disparity | 0.011(0.67) | -0.012(-1.01) | 0.014(0.83) |
| CEO career variety×Cultural difference | 0.069**(2.37) | 0.036***(3.05) | 0.094***(3.21) |
| Firm size | -0.108***(-9.73) | 0.349***(31.58) | -0.175***(-9.04) |
| Leverage | -0.082***(-6.07) | -0.040***(-5.24) | -0.085(-5.56) |
| Growth | 0.100***(3.22) | -0.002(-0.25) | 0.132***(3.97) |
| M&A experience | 0.023(1.26) | -0.034***(-2.76) | 0.018(1.16) |
| Wald chi2 | 964.23*** | 1712.58*** | 886.22*** |

Note: The Z test value is in parentheses.

*Significant at the 10% level, for two-tailed tests.

**Significant at the 5% level, for two-tailed tests.

***Significant at the 1% level, for two-tailed tests.

primarily achieved through the mediation of DKBE, particularly because in the M&A process, geographical distance hinders the efficiency of information communication between the target and acquirer firms, leads to communication complications, reduces the quality of information communication, and increases the negative impact of information asymmetry [56]. From the DKBE perspective, knowledge spillovers have obvious geographical limitations. The greater the intensity of tacit knowledge in the knowledge transfer process, the greater the need for face-to-face communication and contact [57]. Consequently, there is greater geographical distance, which makes digital knowledge transfer more difficult in digital M&A contexts. Therefore, for the geographical distance factor, a greater geographic distance between the M&A parties can hinder the CEO's career variety role and is not conducive to expanding the digital knowledge base or enhancing the degree of DT in digital M&A.

The variable "CEO career variety x Knowledge disparity" is not significant in any of the three models, indicating that knowledge disparity does not moderate the relationship between CEO career variety and DT. The knowledge disparity factor, regardless of the size of knowledge disparity between the target and acquirer firms, does not affect the process of CEO career variety acting on DKBE and the degree of DT. Although scholars have suggested that too large a knowledge stock disparity between the target and acquirer firms may lead to the destruction of the organizational routine of the acquirer [14], digital knowledge differs from general knowledge in that the former is characterized by homogeneity and programmability and can be efficiently disseminated and reorganized across firm boundaries [2]. This lays the foundation for digital knowledge transfer by the target firm in the digital M&A process, resulting in the digital knowledge transfer in the digital M&A context being less affected by knowledge disparity.

The coefficient of "CEO career variety x Cultural difference" is positive in all three models and significant at the 1% or 5% level, indicating that cultural difference has a positive moderating effect on the relationship between CEO career variety and DT. This moderating effect is achieved mainly through the mediating variable DKBE because the degree of digital M&A

integration is reduced compared to other M&A. To ensure the relative stability of the management team and the integrity of digital assets, the M&A parties do not integrate too much into target's core digital R&D and operations, thus greatly reducing the cost burden of cultural differences. At the same time, multicultural integration enables firms with different cultural roots to seek richer cooperation and innovation in multiple dimensions, such as technology and products. In the case of large cultural differences, the acquirer broadens its digital knowledge boundaries and provides creative sources for subsequent digital activities [58]. Therefore, when large cultural differences exist between the target and acquirer firms, the CEO can take full advantage of professional variety to promote DKBE, thereby enhancing DT.

## 5 Discussion

Digital M&A has become a strategic choice for an increasing number of firms to cope with digital disruption and achieve DT and digital development [4]. Although Bughin et al. and Margiono have proposed the preliminary idea of "achieving enterprise DT through digital M&A," which has laid a foundation for subsequent research [18, 19]. There is currently a lack of empirical research on the relationship between the two concepts, and the question of "how to do it" has not yet been resolved, i.e., how to achieve firm DT through digital M&A is still unclear. From the current study of firm DT, we can see that the main internal influences include technical and organizational aspects. According to the current research by Hanelt et al. and Zhou et al. on digital M&A, we find that the process of digital M&A itself includes the flow and transfer of digital resources and other technical aspects [1, 9]. Therefore, we focus on the organizational level and pay attention to the relationship between CEO career variety and firm DT in the context of digital M&A, inspired by the research on managerial individual characteristics and DT [10, 11]. Our study has found that CEO career variety has a positive impact on firm DT, which not only confirms the research by Zhu et al. and Hu et al. but also represents a breakthrough in studying the implementation path of firm DT in the context of digital M&A [1, 9]. In addition, compared to existing research on CEO career variety, we first inherit the basic perspective of high-order theory that CEO career variety can affect their cognitive and behavioral patterns, thereby affecting firm behavior [7, 8]. Subsequently, existing studies have mostly focused on the impact of CEO career variety on corporate governance [8, 33, 34], while we further verify its positive impact on DT based on this foundation, inspiring firms to pay more attention to CEO career experiences and individual characteristics.

In traditional M&A knowledge base research, scholars have mainly focused on the impact of M&A knowledge base integration on market value and innovation performance [12, 59]. Based on the traditional knowledge-based viewpoint, Hanelt et al. further proposed the idea of "expanding digital knowledge bases through digital M&A" [1]. Inspired by Hanelt et al.'s view of digital knowledge bases, this study views DKBE as a unique phenomenon within the context of digital M&A. As tacit knowledge transfer between organizations is difficult, the focus of previous M&A knowledge base research has often been on reducing integration costs to facilitate knowledge transfer [12, 59]. However, the dynamic and scalable nature of digital knowledge allows it to spread and restructure quickly and efficiently across firm boundaries [2]. The characteristics of digital knowledge make it possible to achieve DT through digital M&A. Therefore, this study connects DKBE as a CEO career variety and DT relationship, finding that DKBE plays a mediating effect therein. This finding not only inherits the traditional knowledge-based viewpoint, recognizing the importance of expanding the firm's knowledge base through M&A [13], but also focuses on the important impact of DKBE on DT in the context of digital M&A. It forms a functional relationship that ranges from CEO career variety, DKBE to

DT. This study provides a refined path plan for achieving DT in the context of digital M&A, and provides a new perspective on firm DT.

Through a subsequent heterogeneity analysis, we found significant differences in the moderating effects of three moderating variables—geographic distance, knowledge disparity, and cultural difference—in the original model. A closer geographic distance between the M&A parties facilitates communication, information transfer, and knowledge spillover [56]. Although some scholars argue that the emergence of information technology has weakened geospatial ties between firms, this study verifies that the role of CEO career variety in the digital M&A context is still limited by geographic distance and that geographic distance still has some negative effects on digital knowledge transfer and digital knowledge spillover. The knowledge disparity between M&A parties represents the contrast between the two parties' knowledge stocks. While some have argued that excessive knowledge disparity between target and acquirer firms can lead to the disruption of their organizational routines [14], the results of this study show that the moderating effect of knowledge disparity is not significant, which reaffirms the above view that "the severity of knowledge matching and compatibility problems between the two parties is greatly diminished in the digital M&A field." The research results on the impact of cultural differences on M&A outcomes are not uniform, with some scholars stating that greater cultural differences between M&A parties can increase acquisition costs and inhibit acquisitions. A large cultural difference can make it more difficult to search for a target and lead to higher cultural integration costs [55]. However, others have found that multicultural integration helps firms absorb and integrate diverse cultures, develop diverse products to meet different consumer needs, and thus contribute to their performance after an M&A [60]. The empirical results show that cultural differences have a positive moderating effect, indicating that the presence of cultural differences between the two parties in the digital M&A process helps CEOs play the advantages of professional diversity and promotes DKBE and DT.

## 6 Conclusion and policy suggestion

### 6.1 Conclusion

Using the emerging field of digital M&A as a research context and Chinese NGIT firms as research objects, this study investigates and explores the relationship between CEO career variety, DKBE, and DT. Our results confirm that CEO career variety can promote DKBE and thus enhance the degree of DT, indicating that firms need to pay attention to CEOs' career diversity background when selecting them to improve the quality of CEO digital decisions. In the analysis of heterogeneity influence, geographic distance, knowledge disparity, and cultural differences between target and acquirer firms are introduced as boundary factors, revealing that a longer geographic distance and larger cultural differences have a suppressive and facilitating effect on the above relationship, respectively. However, the boundary effect of knowledge disparity was not significant. This study demonstrates the need to consider the heterogeneous effects of geographical distance and cultural differences between M&A parties in the process of promoting DT through CEO career variety and inspires firms to focus on the selection of target firms in the digital M&A decision-making process.

### 6.2 Policy suggestion

This study offers the following insights for NGIT and other emerging firms motivated by digital M&A and DT. First, it is important for CEOs to pay close attention to career variety. Firms need to emphasize the core position of the CEO in the process of digital M&A, implement sound training and selection mechanisms, and comprehensively consider all aspects of CEOs'

abilities. In turn, CEOs need to focus on the role of rich management practices to improve management ability, not adhere to a certain position or field of work inertia, learn different areas of thinking and management styles, and improve their own knowledge structure and management skills.

Second, regarding the digital knowledge-based expansion process, firms need to attach importance to digital knowledge integration in the process of digital M&A and continuously improve their digital capabilities to ensure effective acquisition, absorption, and utilization of the target party's digital knowledge. Simultaneously, it is necessary to formulate reasonable knowledge integration strategies by combining the specific digital knowledge situations of both M&A parties, including specific choices such as rapid integration, delayed integration, and step-by-step integration.

Finally, appropriate target firm selection is important. Geographical distance and cultural differences can influence the final DT effect, and acquirer firms should take advantage of multiculturalism to promote digital knowledge transfer and applications. Acquirer firms must also consider and evaluate the impact of geographic distance on M&A integration when searching for and screening potential firms as M&A targets. When geographic distance is unavoidable, its negative impact is mitigated by the subsequent enhancement of management communication and information transfer.

## Supporting information

**S1 File.**
(ZIP)

## Author Contributions

**Conceptualization:** Hongyang Li.

**Data curation:** Xu Yang.

**Methodology:** Xu Yang, Mingming Meng.

**Validation:** Xu Yang.

**Writing – original draft:** Hongyang Li.

**Writing – review & editing:** Mingming Meng.

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
