## [Decision Letter · Decision Letter 0]

6 Sep 2023

PONE-D-23-24389Research on the relationship between CEO career variety, digital knowledge base extension, and digital transformation in the context of digital merger and acquisition: The case of China's new generation of information technology firmsPLOS ONE

Dear Dr. Li,

Thank you for submitting your manuscript to PLOS ONE. After careful consideration, we feel that it has merit but does not fully meet PLOS ONE’s publication criteria as it currently stands. Therefore, we invite you to submit a revised version of the manuscript that addresses the points raised during the review process.

We look forward to receiving your revised manuscript.

Kind regards,

José Antonio Clemente Almendros, PhD

Academic Editor

PLOS ONE

Journal Requirements:

 "This work was supported by the National Natural Science Foundation of China [grant number 72274044]."  

**Additional Editor Comments:**

Dear Authors.

I believe that your study is very promising. However, I also agree with the reviewers about the need of major improvements. Additionally to the reviewers' opinion, I encourage you to proceed with the following issues:

-  INTRODUCTION: I miss the contribution of your study

-  RESUTS: I miss a table with the description of the variables, how they have been calculated, .... Regarding the contructs from factor analysis, I miss the related parameters (such as Alpha Cronbach, KMO,...). I also miss a correlation table. I am concern about the potential endogeneity of your model, please the study needs to include how you cope with this potential issue.

- DISCUSSION: I encourage you to show in this section what is the new knowledge your study add to the related literature 

Reviewers' comments:

Reviewer's Responses to Questions

**Comments to the Author**

1. Is the manuscript technically sound, and do the data support the conclusions?

Reviewer #1: Yes

Reviewer #2: Yes

2. Has the statistical analysis been performed appropriately and rigorously? 

Reviewer #1: Yes

Reviewer #2: Yes

3. Have the authors made all data underlying the findings in their manuscript fully available?

Reviewer #1: Yes

Reviewer #2: Yes

4. Is the manuscript presented in an intelligible fashion and written in standard English?

Reviewer #1: Yes

Reviewer #2: Yes

5. Review Comments to the Author

Reviewer #1: Thank you for giving me the opportunity to review this manuscript. The topic is of interest and the research is sound. There are few aspects to be improved:

1- the literature review of this topics are very broad and it is not reflected in the paper. I suggest to develop a subsection for the main variables studied (digital transformation, knowledge disparities...)

2. table 1 - there is 2 times Knowledge disparities and missing cultural differences - please fix it.

3. A figure with the research framework and finding will increase the readability

4. I expect the discussions and conclusions should be presented toward the other studies from the literature.

Reviewer #2: Although the study is a very interesting study, it is suggested to be improved in some points. These points are explained below:

*Why was the stepwise regression method chosen? This part should be explained in detail.

*Tables should be standardized.

*The endogeneity problem is not addressed in these models. It should at least be included and discussed in the robustness checks section.

*Measurement of variables and how they are calculated should be tabulated in more detail.

*The connection of the study with other studies should be given under a discussion section.

*Diagnostic tests of regression analysis should also be included. R2 should be given at least.

All the best,

6. PLOS authors have the option to publish the peer review history of their article (what does this mean?). If published, this will include your full peer review and any attached files.

Reviewer #1: No

Reviewer #2: No

---

## [Author Response · Author response to Decision Letter 0]

9 Nov 2023

October 18, 2023

Re: Response for manuscript PONE-D-23-24389 “Research on the relationship between CEO career variety, digital knowledge base extension, and digital transformation in the context of digital merger and acquisition: The case of China's new generation of information technology firms”

Dear Editor Almendros,

Thanks for providing us with this great opportunity to submit a revised version of our manuscript. We appreciate the detailed and constructive comments provided by the reviewers. We have carefully revised the manuscript by incorporating all the suggestions by the review panel.

Appended to this letter is our point-by-point response to the comments raised by the reviewers. The comments are reproduced and our responses are given directly afterward in a different color (red).

We hope this revised manuscript has addressed your concerns, and look forward to hearing from you.

Sincerely,

Hongyang Li

SUGGESTIONS FROM EDITOR

Comment 1: Please ensure that your manuscript meets PLOS ONE's style requirements, including those for file naming. 

Response 1: Thank you for the detailed review. We have consulted the official website of the PLOS ONE journals to ensure that our manuscript conforms to the required style. We have also referred to the format of recently published papers in the journals and made comprehensive modifications and improvements to the manuscript accordingly. These include adjustments to the chapter structure, changes to table formatting, and updates to the reference format.

Comment 2: Please ensure that you include a title page within your main document. We do appreciate that you have a title page document uploaded as a separate file, however, as per our author guidelines we do require this to be part of the manuscript file itself and not uploaded separately.

Response 2: Thank you for the detailed review. We have included the title page at the beginning of the manuscript as required, and listed all authors and their affiliations. In particular, we have uploaded the document "Changes to Authorship" and applied to the editorial department for a change in the order of the first and second authors. This involved a change in the corresponding author from Hongyang Li from the second author to the first author. We provided the reason for this change and the electronic signatures of all three authors in the document. Therefore, we have temporarily listed the changed order of authors on the title page, pending further approval from the editorial department. We hope that the editorial department will carefully consider our request.

Comment 3: In your Data Availability statement, you have not specified where the minimal data set underlying the results described in your manuscript can be found.

Response 3: Thank you for the detailed review. We are going to include the data compression package in the "Supporting information" section of the manuscript, and have uploaded the compression package in the attachment, which includes the data set and the running command.

Comment 4: Thank you for stating the following financial disclosure: "This work was supported by the National Natural Science Foundation of China [grant number 72274044]."

Response 4: Thank you for the detailed review. We need to report to the editorial department, the project funders are not among the three authors, and we strongly agree with the financial disclosure suggestions given by the editorial department: "The funders had no role in study design, data collection and analysis, decision to publish, or preparation of the manuscript."

SUGGESTIONS FROM ADDITIONAL EDITOR

Comment 1: INTRODUCTION: I miss the contribution of your study.

Response 1: Thanks for your great suggestion on improving the contribution of our manuscript. We fully support the editor's suggested changes. We have moved the theoretical contribution from the "Conclusion" section to the "Introduction" section, and have adjusted and integrated the paragraphs of the introduction to make it more logical and refine the language.

Comment 2: RESUTS: I miss a table with the description of the variables, how they have been calculated, .... Regarding the contructs from factor analysis, I miss the related parameters (such as Alpha Cronbach, KMO,...). I also miss a correlation table. I am concern about the potential endogeneity of your model, please the study needs to include how you cope with this potential issue.

Response 2: Thanks for your great suggestion on improving the standardization of our manuscript. 

(1)We have added a more specific description of factor analysis in the "Dependent variable" section of Chapter 3. Firstly, we have included a description of the step-by-step process of factor analysis. Secondly, we have added the formula for calculating the composite score in the third step, along with an explanation of the variables involved in the formula. Finally, we have listed the calculation results of factor analysis, including the KMO value, chi-square value of Bartlett's sphericity, and core indicators such as significance, common factor extraction degree, and cumulative variance contribution rate. The new parts in the manuscript are as follows:ce contribution rate. The new parts in the manuscript are as follows:

To avoid the multicollinearity issue, we employed the factor analysis method to reduce the dimensionality of the aforementioned indicators and computed the comprehensive score for the CEO career experience richness index through weighting. The first step involved standardizing the five indicators and calculating the KMO value as well as the Bartlett test of sphericity to determine the applicability of the factor analysis method. Subsequently, in the second step, we extracted the common factor based on the criterion of an eigenvalue greater than one and calculated the score of the common factor in different samples. Finally, in the third step, we constructed a comprehensive evaluation function that relied on the common factor score and the variance contribution rate of this factor to obtain a comprehensive score. The specific calculation formula is as follows

Where Indexti represents the comprehensive score of the CEO career experience richness index for the ith sample company in year t; Fij represents the extracted common factor; and βij represents the variance contribution rate of the j-th common factor for the i-th sample company.

After calculation, the KMO value of the samples is 0.716, and the chi-square value of Bartlett’s sphericity is 270.465, which is significant at the level of 1%, indicating that the samples are suitable for factor analysis. The common factor extraction degrees are all above 0.5, and the extraction degree of the original variables is relatively high. The common factors are extracted according to the standard that the characteristic root is greater than 1, and after rotation by the maximum variance method, the cumulative variance contribution rate is 62.044%, which can better represent most information of the original variables. Finally, the comprehensive score of CEO’s career experience richness index is calculated based on Formula (1).

(2)We add the correlation analysis in Section 4.1 "Descriptive statistics and correlation analysis," and list the correlation analysis results and VIF values among variables in Table 3. The new contents and tables in the manuscript are as follows:

Table 3 shows that the correlation coefficients of the core variables are all less than 0.3, and the VIFs fall below the standard of 10, indicating the absence of multicollinearity issues. Furthermore, we observed that there is generally no linear correlation between the core variables, which suggests the possibility of non-linear relationships within the data as well as heteroscedasticity issues that merit further investigation in subsequent analyses.

(3)We would like to address some concerns regarding the potential endogeneity in the model. After careful consideration, we provide the following explanations and supplements.

First and foremost, we need to elucidate the endogeneity issue in the context of M&A. Endogeneity problems that may exist in empirical studies and require verification include backward causation and selection bias. In general empirical studies, it is typically necessary to employ the instrumental variable method to address the backward causation issue, or utilize the Heckman two-stage method to address the selection bias problem.

Secondly, there is typically no backward causation issue in M&A research, as M&A research typically has a distinct division between the period before and after the merger. Taking this study as an example, the dependent variable in this study is CEO career variety, which represents the characteristics of the CEO prior to the occurrence of the M&A. The mediating variable is DKBE, which represents the aggregate value of digital patents lagged three years by M&A. The independent variable is DT, which represents the degree of DT in the fourth year after M&A. It can be inferred that the process from CEO career variety to DKBE to DT follows an evident timeline. Therefore, in this case, there is generally no backward causation problem, that is, the degree of DT four years after M&A will not adversely affect CEO career variety four years before M&A. Our view can also be verified by consulting well-known M&A literature in recent years. Both digital M&A and general M&A research, such as Hanelt et al. (2021), McCarthy and Aalbers (2016), Bauer et al. (2016), Cheng and Yang(2017), Shafique and Hagedoorn et al. (2022), did not conduct endogeneity tests.

Finally, we address the issue of selection bias in the endogeneity problem. The characteristics of companies that choose M&A may differ from those that do not, that is to say, whether an enterprise chooses M&A is not a random event (Ahern et al., 2015). Therefore, there may be selection bias in M&A research. Selection bias means that the selection of samples is non-random and may be samples of a specific population, which may not represent the population (Hill et al., 2021). In practical research, this problem is inevitable, and it is only possible to test whether the selection bias problem has a noticeable impact on the results through the Heckman two-stage method. In M&A research, many scholars also use this method as a robustness test or endogeneity test (Sears and Hoetker, 2014; Thakur-Wernz et al., 2019; Zhou et al., 2021). In this study, the Heckman two-stage method is used to conduct an endogeneity test, and the test is carried out by referring to the steps of Ahern et al. (2015). The endogeneity test added to the manuscript is as follows:

The characteristics of firms that choose digital M&A s may not be the same as those that do not; that is, whether a firm chooses M&A is not a random event [55]. To avoid bias in the estimation results, the Heckman two-stage method is used to test for endogeneity. First, we select a sample of new generation information technology listed companies from 2013 to 2017 and conduct a panel probit with whether the company conducted digital M&A in the current year as the explanatory variables, and a total of eight indicators, including firm size, total asset turnover, total asset return, return on net assets, gearing ratio, growth, number of technicians, and amount of R&D investment in the year before M&A as explanatory variables. The regression, and the Inverse Mills Ratio are calculated. Second, we put the Inverse Mills Ratio into the mediation test model as a control variable. As shown in Table 6, the Inverse Mills Ratio values in the model are significant, indicating that there is a certain selection bias issue in the model. However, after controlling for Inverse Mills Ratio, the core variables in all models do not undergo substantial changes, indicating that the selection bias issue has a relatively small impact on the results and the study findings remain robust.

The references involved in response 2:

Ahern, K. R., Daminelli, D., & Fracassi, C. (2015). Lost in translation? The effect of cultural values on mergers around the world. Journal of Financial Economics, 117(1), 165-189.

Bauer, F., Matzler, K., & Wolf, S. (2016). M&A and innovation: The role of integration and cultural differences—A central European targets perspective. International Business Review, 25(1), 76-86.

Cheng, C., & Yang, M. (2017). Enhancing performance of cross-border mergers and acquisitions in developed markets: The role of business ties and technological innovation capability. Journal of Business Research, 81, 107-117.

Hanelt, A., Firk, S., Hildebrandt, B., & Kolbe, L. M. (2021). Digital M&A, digital innovation, and firm performance: an empirical investigation. European Journal of Information Systems, 30(1), 3-26.

Hill, A. D., Johnson, S. G., Greco, L. M., O’Boyle, E. H., & Walter, S. L. (2021). Endogeneity: A review and agenda for the methodology-practice divide affecting micro and macro research. Journal of Management, 47(1), 105-143.

McCarthy, K. J., & Aalbers, H. L. (2016). Technological acquisitions: The impact of geography on post-acquisition innovative performance. Research Policy, 45(9), 1818-1832.

Sears, J., & Hoetker, G. Technological overlap, technological capabilities, and resource recombination in technological acquisitions. Strategic Management Journal. 2014; 35(1), 48-67.

Shafique, M., & Hagedoorn, J. (2022). Look at U: Technological scope of the acquirer, technological complementarity with the target, and post-acquisition R&D output. Technovation, 115, 102533.

Thakur-Wernz, P., Cantwell, J., & Samant, S. (2019). Impact of international entry choices on the nature and type of innovation: Evidence from emerging economy firms from the Indian bio-pharmaceutical industry. International Business Review, 28(6), 101601.

Zhou, J., Liu, C., Xing, X., & Li, J. (2021). How can digital technology-related acquisitions affect a firm’s innovation performance?. International Journal of Technology Management, 87(2-4), 254-283.

Comment 3: DISCUSSION: I encourage you to show in this section what is the new knowledge your study add to the related literature

Response 3: Thanks for your great suggestion on the logic of our manuscript. Through our reading of the manuscript, we identified several shortcomings in the discussion section. We have re-analyzed and rewritten the discussion section, reviewed relevant literature, connected the gaps in previous literature with the contributions of our paper, and fully evaluated the inheritance and development of similar literature from a theoretical perspective. The first part delves into the relationship between CEO career diversity and digital transformation in the context of digital M&A; the second part explores the mediating role of DKBE; and the third part examines the moderating variables. The specific new additions are as follows (the additions are marked in red):

Digital M&A has become a strategic choice for an increasing number of firms to cope with digital disruption and achieve DT and digital development [4]. Although Bughin et al. and Margiono have proposed the preliminary idea of "achieving enterprise DT through digital M&A," which has laid a foundation for subsequent research [18,19]. There is currently a lack of empirical research on the relationship between the two concepts, and the question of "how to do it" has not yet been resolved, i.e., how to achieve firm DT through digital M&A is still unclear. From the current study of firm DT, we can see that the main internal influences include technical and organizational aspects. According to the current research by Hanelt et al. and Zhou et al. on digital M&A, we find that the process of digital M&A itself includes the flow and transfer of digital resources and other technical aspects [1,9]. Therefore, we focus on the organizational level and pay attention to the relationship between CEO career variety and firm DT in the context of digital M&A, inspired by the research on managerial individual characteristics and DT [10,11]. Our study has found that CEO career variety has a positive impact on firm DT, which not only confirms the research by Zhu et al. and Hu et al. but also represents a breakthrough in studying the implementation path of firm DT in the context of digital M&A [1,9]. In addition, compared to existing research on CEO career variety, we first inherit the basic perspective of high-order theory that CEO career variety can affect their cognitive and behavioral patterns, thereby affecting firm behavior [7,8]. Subsequently, existing studies have mostly focused on the impact of CEO career variety on corporate governance [8,33,34], while we further verify its positive impact on DT based on this foundation, inspiring firms to pay more attention to CEO career experiences and individual characteristics.

In traditional M&A knowledge base research, scholars have mainly focused on the impact of M&A knowledge base integration on market value and innovation performance [12,56]. Based on the traditional knowledge-based viewpoint, Hanelt et al. further proposed the idea of "expanding digital knowledge bases through digital M&A" [1]. Inspired by Hanelt et al.’s view of digital knowledge bases, this study views DKBE as a unique phenomenon within the context of digital M&A. As tacit knowledge transfer between organizations is difficult, the focus of previous M&A knowledge base research has often been on reducing integration costs to facilitate knowledge transfer [12,56]. However, the dynamic and scalable nature of digital knowledge allows it to spread and restructure quickly and efficiently across firm boundaries [2]. The characteristics of digital knowledge make it possible to achieve DT through digital M&A. Therefore, this study connects DKBE as a CEO career variety and DT relationship, finding that DKBE plays a mediating effect therein. This finding not only inherits the traditional knowledge-based viewpoint, recognizing the importance of expanding the firm’s knowledge base through M&A [13], but also focuses on the important impact of DKBE on DT in the context of digital M&A. It forms a functional relationship that ranges from CEO career variety, DKBE to DT. This study provides a refined path plan for achieving DT in the context of digital M&A, and provides a new perspective on firm DT.

Through a subsequent heterogeneity analysis, we found significant differences in the moderating effects of three moderating variables—geographic distance, knowledge disparity, and cultural difference—in the original model. A closer geographic distance between the M&A parties facilitates communication, information transfer, and knowledge spillover [57]. Although some scholars argue that the emergence of information technology has weakened geospatial ties between firms, this study verifies that the role of CEO career variety in the digital M&A context is still limited by geographic distance and that geographic distance still has some negative effects on digital knowledge transfer and digital knowledge spillover. The knowledge disparity between M&A parties represents the contrast between the two parties' knowledge stocks. While some have argued that excessive knowledge disparity between target and acquirer firms can lead to the disruption of their organizational routines [14], the results of this study show that the moderating effect of knowledge disparity is not significant, which reaffirms the above view that “the severity of knowledge matching and compatibility problems between the two parties is greatly diminished in the digital M&A field.” The research results on the impact of cultural differences on M&A outcomes are not uniform, with some scholars stating that greater cultural differences between M&A parties can increase acquisition costs and inhibit acquisitions. A large cultural difference can make it more difficult to search for a target and lead to higher cultural integration costs [55]. However, others have found that multicultural integration helps firms absorb and integrate diverse cultures, develop diverse products to meet different consumer needs, and thus contribute to their performance after an M&A [58]. The empirical results show that cultural differences have a positive moderating effect, indicating that the presence of cultural differences between the two parties in the digital M&A process helps CEOs play the advantages of professional diversity and promotes DKBE and DT.

COMMENTS TO THE AUTHOR:

Reviewer #1:

Comment 1: The literature review of this topics are very broad and it is not reflected in the paper. I suggest to develop a subsection for the main variables studied (digital transformation, knowledge disparities...)

Response 1: Thanks for your great suggestion on improving the literature review of our manuscript. We found that the literature review section lacks clarity. We fully agree with the reviewer that the literature review should be conducted according to the core variables. After combing the literature, we selected three keywords for review: digital M&A, DT, and CEO career variety.

The first keyword is digital M&A, which is an important context for research and also the theoretical foundation of our study. Therefore, it is of great significance to review the research related to digital M&A. The second keyword is DT, which is the independent variable and the core theme of our study. This review of the influencing factors of DT is helpful to clarify the process of DT in non-M&A situations, which can be used as an important reference for digital M&A situations. The third keyword is CEO career variety. CEO career variety is the dependent variable of the study, which affects DT in the context of digital M&A. The review of CEO career variety is conducive to clarifying the specific process of CEO career variety affecting corporate decision-making behavior, and provides reference for the study of DT decision-making. It should be noted that the DKBE variable serves as a mediating variable and is not included in the scope of the review. The reason is that although DKBE is also a core variable of this study, it is only an intermediate process from the role of CEO career diversity to the DT of enterprises. Only discussing the literature related to knowledge base or digital knowledge base cannot well find the current research gap. Therefore, this study puts the introduction of digital knowledge base in the research hypothesis and discussion section, and discusses it in combination with the actual situation of this study. The new parts in the literature review are as follows (marked in red) :

The concept of digital M&A s is based on early research on technological M&A in the digital economy. Since Ahuja and Katila [14] introduced the concept of technology M&A in 2001, technology M&A research has developed rapidly, research on digital M&A is still in its infancy. Although some scholars have focused on general M&A research in the context of the digital economy [15,16,17], their focus differs somewhat from digital M&A. The term digital M&A was first introduced in a research note published by Bain and Company. Subsequently, the idea of “firm DT can be achieved through digital M&A” began to attract scholars’ attention [18,19]. However, to date, studies have focused only on the theoretical level of digital M&A. A search of the Web of Science database reveals only a handful of empirical studies on digital M&A. One such study is Hanelt et al. [1], who through systematic elaboration, theoretical conceptualization, and empirical testing of digital M&A, argued that digital M&A helps promote digital innovation and boosts firm performance. Additionally, Zhou et al. [9], using data of Chinese listed firms as a sample, verified the positive relationship between digital M&A and innovation performance. Yu and Yan [20] argued that digital finance development promotes the implementation of digital M&A by firms. Finally, Tang et al. [21] focused on the positive impact of digital M&A on the market value of Chinese listed firms. However, there is still a lack of research on how to achieve the goal of firm DT through digital M&A.

In the field of influencing factors of firm DT, early scholars focused on technical aspects such as digital resources and digital-related abilities [3,5]. However, as the internal obstacles to DT within firms continue to increase, the organizational element is gradually gaining attention. Scholars have investigated the role of typical organizational factors such as organizational structure, organizational culture, and organizational governance can influence firm DT [3,22], and in this process, the role of managers has gradually emerged. Previous studies have delved into two aspects: manager characteristics and manager abilities, and found that both psychological characteristics, cognitive structures, overseas backgrounds [10,11,23], and digital literacy, digital self-efficacy, and management abilities [24,25,26] have significant impacts on DT. Additionally, the impact of the Chief Digital Officer, a specialized senior management position, on DT has also received attention [27,28].

As the main decision-makers in the daily management of firms, CEOs have been the focus of attention in business and strategic management research. Traditional agency theory focuses on how to guide managers to make "pareto optimality" decisions to promote sustainable development, but the implied premise of managerial homogeneity does not easily match the real situation and is gaining increasingly more attention from scholars. Based on the upper echelons theory, CEOs, as the core of management, have career experiences that influence their cognitive and behavioral patterns, which, in turn, influences firms’ behavior. Previous research on CEO career variety has focused on the economic impacts of single specific career experiences such as military, R&D, and financial experience on firms [29,30,31]. However, different career types interact with each other to shape the management style of the CEO. Some studies have found that CEOs employ a combination of skills learned throughout their careers when making corporate decisions, and that CEOs with a wide range of career experiences are usually more strained, boundary-spanning, innovative, and adventurous [8,32]. However, current research has centered solely on the impact of CEO career variety on outcomes such as CEO compensation, investment decisions, and board tenure [8,33,34], and there is a lack of in-depth exploration into the consequences resulting from CEO career variety.

Comment 2: Table 1 - there is 2 times Knowledge disparities and missing cultural differences - please fix it.

Response 2: We are very sorry for the errors in the manuscript and thank the reviewers for their careful corrections. We corrected the errors mentioned by experts in Table1 (now Table2) and carefully checked other data.

Comment 3: A figure with the research framework and finding will increase the readability

Response 3: Thanks for your great suggestion on improving the readability of our manuscript. After examination, we only have four tables in the original manuscript, so we highly approve the reviewers' suggestions on adding pictures and tables in the manuscript. In this revision process, we added three additional tables and four formulas, so as to more clearly reflect the data results of the article. The new tables and formulas are (only the table names are listed here because the new tables are listed elsewhere in the modification instructions) :

Table 1. Descriptive statistics of the variables

Table 3. Correlation analysis results

Table 6. Endogeneity test results

At the same time, we have carefully evaluated the reviewer's suggestion of adding research frame pictures to the manuscript. We believe that adding research frame pictures can indeed more clearly express the relationship between various variables, but we are also worried that the manuscript may not be suitable for adding research frame pictures, the main reasons are as follows: First, the research framework is generally placed in the part of research hypotheses, but the relationship between CEO career diversity, DKBE and digital transformation in this study is very clear, which is directly reflected in the title. At the same time, the three variables have no distinction such as dimension division, so it is not necessary to add the research framework picture of the connection between the three variables in the research hypothesis part. Second, we include three moderating variables in our further analysis. However, from the perspective of overall logic, this part is not suitable to appear in the research hypothesis in advance. We also considered whether we could put the research framework diagram into the discussion section, but again the necessity was not large enough, and no additional information needed to be presented more clearly through pictures.

Based on the above considerations, we once again thank the reviewers for their suggestions and decide not to add the research framework diagram for the time being. We will also continue to think about how to better present the manuscript point of view and express the legibility of the manuscript more clearly.

Comment 4: I expect the discussions and conclusions should be presented toward the other studies from the literature.

Response 4: Thanks for your great suggestion on improving the discussion section of our manuscript. We found that both the editors and the reviewers made suggestions to further modify the discussion section and strengthen the connection with other literature. We have already mentioned the changes to the discussion section in response 3 to the Additional Editor's suggestion, and shown the changes in the manuscript. Thanks again to the reviewers for their suggestions.

Reviewer #2:

Comment 1: Why was the stepwise regression method chosen? This part should be explained in detail.

Response 1: Thanks for your great suggestion on improving the readability of our manuscript. We find that the introduction of stepwise regression is indeed neglected in the manuscript, and the part about model building is also missing in the manuscript. Therefore, we add Section 3.3 Model to strengthen the necessity of using the stepwise regression method to verify the mediation problem by quoting the literature of Baron and Kenny (1986) and Hayes (2009). At the same time, due to the characteristics of the secondary data in this study, the mediating effect test methods such as structural equation model and bootstrap are not applicable to this study. Through the inquiry of other similar studies, the stepwise regression method is also widely used to solve the intermediary problem in the M&A theme, which will not be further described here, such as Hault et al. (2021), Zhou et al. (2021). Section 3.3 is added as follows:

The stepwise testing procedure proposed by Baron and Kenny [53] is currently regarded as the most popular method for testing the mediating effect. Baron and Kenny [53] argue that the test of the mediating effect should involve three steps: Firstly, testing the total effect of the independent variable on the dependent variable (i.e., whether β1 in Equation 2 is significant); Secondly, testing the effect of the independent variable on the proposed mediator (i.e., whether β1 in Equation 3 is significant); Thirdly, testing the effect of the proposed mediator on the dependent variable controlling for the independent variable (i.e., whether β2 in Equation 4 is significant). Hayes [54] also refers to this process as the causal steps approach or joint significance test.

The references involved in response 1:

Baron, R. M., & Kenny, D. A. (1986). The moderator-mediator variable distinction in social psychological research: Conceptual, strategic, and statistical considerations. Journal of personality and social psychology, 51(6), 1173–1182. 

Hanelt, A., Firk, S., Hildebrandt, B., & Kolbe, L. M. (2021). Digital M&A, digital innovation, and firm performance: an empirical investigation. European Journal of Information Systems, 30(1), 3-26.

Hayes, A. F. (2009). Beyond Baron and Kenny: Statistical mediation analysis in the new millennium. Communication monographs, 76(4), 408-420. 

Zhou, J., Liu, C., Xing, X., & Li, J. (2021). How can digital technology-related acquisitions affect a firm’s innovation performance?. International Journal of Technology Management, 87(2-4), 254-283.

Comment 2: Tables should be standardized.

Response 2: Thanks for your great suggestion on improving the rigor of our manuscript. Our manuscript does not meet the requirements of the journal in some aspects of style and format. We have referred to the official website of the journal to modify the table presentation of the manuscript, including the border of the table and the issue of bold font in the table. We also carefully checked other typesetting problems in the manuscript, and made modifications and improvements.

Comment 3: The endogeneity problem is not addressed in these models. It should at least be included and discussed in the robustness checks section.

Response 3: Thanks for your great suggestion on improving the robustness of our manuscript. We found that both editors and reviewers raised concerns about the endogeneity test, which our manuscript did neglect. Fortunately, after repeated reminders from editors and reviewers, we gradually paid attention to the endogeneity problem in our research. We have explained the detailed modification theory and modification process in response 2 to the suggestions of Additional Editor, and added the section of endogeneity test in the manuscript.

Comment 4: Measurement of variables and how they are calculated should be tabulated in more detail.

Response 4: Thanks for your great suggestion on improving the readability of our manuscript. We summarized the variable measurement section in Table 1 as suggested by the reviewers. Through modification, we find that the measurement process of each variable can be seen more clearly through the table. Thank you again for your valuable suggestions. 

Comment 5: The connection of the study with other studies should be given under a discussion section.

Response 5: Thanks for your great suggestion on improving the discussion section of our manuscript. After careful discussion, we modified and improved the discussion section. We have already mentioned the changes to the discussion section in response 3 to the Additional Editor's suggestion, and shown the changes in the manuscript. Thanks again to the reviewers for their suggestions.

Comment 6: Diagnostic tests of regression analysis should also be included. R2 should be given at least.

Response 6: Thanks for your great suggestion on improving the regression analysis section of our manuscript. We need to explain and explain to the reviewers. The model used in the regression analysis of this study is feasible generalized least squares (FGLS), which is quite different from the conventional OLS model because there is no R-squared statistic in the FGLS model.

The R-squared statistic is an ordinary least squares (OLS) concept that is useful because of the unique way it breaks down the total sum of squares into the sum of the model sum of squares and the residual sum of squares. When you estimate the model’s parameters using feasible generalized least squares (FGLS), the total sum of squares cannot be broken down in the the same way, making the R-squared statistic less useful as a diagnostic tool for FGLS regressions. Specifically, an R-squared statistic computed from FGLS sums of squares need not be bounded between zero and one and does not represent the percentage of total variation in the dependent variable that is accounted for by the model. Also, eliminating or adding variables in a model does not always increase or decrease the computed R-squared value.

Wald test is commonly used to test the significance of the FGLS model, and chi-squared statistics are employed to assess the model's robustness. The Wald test bears some similarities to, but also several differences from, the F test in the OLS model. Similar to the F-test, the Wald test is a test of several constrained hypotheses. However, unlike the F-test, which requires the random disturbance term to satisfy the normal distribution and is a hypothesis test for linear constraints, the Wald test is applicable when no assumptions are made about the distribution and when the constraints are nonlinear. In such cases, a Wald test is necessary to reject the null hypothesis that each estimator is equal to zero, in order to demonstrate the significance of the entire model.

In practice, the Wald test and F test are usually judged by STATA software based on the specified model, and the chi-squared statistics in the Wald test and the F value in the F test can be calculated mutually. We refer to the explanation of F and chi-squared statistics in the STATA software operation instructions:

F and chi-squared statistics are really the same thing in that, after a normalization, chi-squared is the limiting distribution of the F as the denominator degrees of freedom goes to infinity. The normalization is:

chi-squared = (numerator degrees of freedom) * F

Therefore, it can be found that the R-squared statistic is not applicable to the FGLS model, and the significance of the model needs to be judged by the Wald chi2 value (rather than the F value). Wu et al. (2022) and Chen (2023) compared OLS and FGLS results, in which R-squared statistic was used in OLS results. However, the R-squared statistic is not used in the FGLS results of the above literature, but the Wald test is used to judge the robustness of the model. Other literature using the FGLS model can also prove the above point (Lindstrom et al. 2020; Islam et al. 2021).

The references involved in response 6:

Chen, H. (2023). Energy innovations, natural resource abundance, urbanization, and environmental sustainability in the post-covid era. Does environmental regulation matter?. Resources Policy, 85, 103882.

Islam, M. A., Liu, H., Khan, M. A., Islam, M. T., & Sultanuzzaman, M. R. (2021). Does foreign direct investment deepen the financial system in Southeast Asian economies?. Journal of Multinational Financial Management, 61, 100682.

Lindström, H., Lundberg, S., & Marklund, P. O. (2020). How Green Public Procurement can drive conversion of farmland: An empirical analysis of an organic food policy. Ecological Economics, 172, 106622.

Wu, H., Ba, N., Ren, S., Xu, L., Chai, J., Irfan, M., ... & Lu, Z. N. (2022). The impact of internet development on the health of Chinese residents: transmission mechanisms and empirical tests. Socio-Economic Planning Sciences, 81, 101178.

---

## [Decision Letter · Decision Letter 1]

28 Dec 2023

Research on the relationship between CEO career variety, digital knowledge base extension, and digital transformation in the context of digital merger and acquisition: The case of China's new generation of information technology firms

PONE-D-23-24389R1

Dear Dr. Li,

We’re pleased to inform you that your manuscript has been judged scientifically suitable for publication and will be formally accepted for publication once it meets all outstanding technical requirements.

Kind regards,

José Antonio Clemente Almendros, PhD

Academic Editor

PLOS ONE

Additional Editor Comments (optional):

Reviewers' comments:

Reviewer's Responses to Questions

**Comments to the Author**

1. If the authors have adequately addressed your comments raised in a previous round of review and you feel that this manuscript is now acceptable for publication, you may indicate that here to bypass the “Comments to the Author” section, enter your conflict of interest statement in the “Confidential to Editor” section, and submit your "Accept" recommendation.

Reviewer #1: All comments have been addressed

Reviewer #2: All comments have been addressed

2. Is the manuscript technically sound, and do the data support the conclusions?

Reviewer #1: Yes

Reviewer #2: Yes

3. Has the statistical analysis been performed appropriately and rigorously? 

Reviewer #1: Yes

Reviewer #2: Yes

4. Have the authors made all data underlying the findings in their manuscript fully available?

Reviewer #1: Yes

Reviewer #2: Yes

5. Is the manuscript presented in an intelligible fashion and written in standard English?

Reviewer #1: Yes

Reviewer #2: Yes

6. Review Comments to the Author

Reviewer #1: Thank you for considering my recommendations and proceed with the implementation.

I am satisfied with the improvements made.

Good luck!

Reviewer #2: (No Response)

7. PLOS authors have the option to publish the peer review history of their article (what does this mean?). If published, this will include your full peer review and any attached files.

Reviewer #1: No

Reviewer #2: No

---

## [Editor Report · Acceptance letter]

29 Jan 2024

PONE-D-23-24389R1 

PLOS ONE

Dear Dr. Li, 

I'm pleased to inform you that your manuscript has been deemed suitable for publication in PLOS ONE. Congratulations! Your manuscript is now being handed over to our production team.

Kind regards, 

on behalf of

Dr. José Antonio Clemente Almendros 

Academic Editor

PLOS ONE